# TEMPORAL DEPENDENCIES IN FEATURE IMPORTANCE FOR TIME SERIES PREDICTION

**Kin Kwan Leung**  **Clayton Rooke**  **Jonathan Smith**  **Saba Zuberi**  **Maksims Volkovs**
Layer 6 AI       Univ. Waterloo    Meta             Layer 6 AI      Layer 6 AI

## ABSTRACT

Time series data introduces two key challenges for explainability methods: firstly, observations of the same feature over subsequent time steps are not independent, and secondly, the same feature can have varying importance to model predictions over time. In this paper, we propose Windowed Feature Importance in Time (WinIT), a feature removal based explainability approach to address these issues. Unlike existing feature removal explanation methods, WinIT explicitly accounts for the temporal dependence between different observations of the same feature in the construction of its importance score. Furthermore, WinIT captures the varying importance of a feature over time, by summarizing its importance over a window of past time steps. We conduct an extensive empirical study on synthetic and real-world data, compare against a wide range of leading explainability methods, and explore the impact of various evaluation strategies. Our results show that WinIT achieves significant gains over existing methods, with more consistent performance across different evaluation metrics.

## 1 INTRODUCTION

Reliably explaining predictions of machine learning models is important given their wide-spread use. Explanations provide transparency and aid reliable decision making, especially in domains such as finance and healthcare, where explainability is often an ethical and legal requirement (Amann et al., 2020; Prenio & Yong, 2021). Multivariate time series data is ubiquitous in these sensitive domains, however explaining time series models has been relatively under explored. In this work we focus on saliency methods, a common approach to explainability that provides explanations by highlighting the importance of input features to model predictions (Baehrens et al., 2010; Mohseni et al., 2020).

It has been shown that standard saliency methods underperform on deep learning models used in the time series domain (Ismail et al., 2020). In time series data, observations of the same feature at different points in time are typically related and their order matters. Methods that aim to highlight important observations but treat them as independent face significant limitations. Furthermore, it is important to note that the same observation of a feature can vary in importance to predictions at different times, which we refer to as the temporal dependency in feature importance. For example, there can be a delay between important feature shifts and a change in the model's predictions. These temporal dynamics can be difficult for current explainability methods to capture.

To address these challenges, we propose the **Win**dowed Feature **I**mportance in **T**ime (WinIT), a feature removal based method which determines the importance of a given observation to the predictions over a time window. Feature removal-based methods generate importance by measuring the change in outcomes when a particular feature is removed. However, removing a feature at a particular time does not account for the dependence between current and future observations of the same feature. WinIT addresses this by assigning the importance of a feature at a specific time based on a *difference* of two scores, that each take the temporal dependence of the subsequent observations into account. To capture the varying importance of the same feature observation to predictions at different times, WinIT aggregates the importance to predictions over a window of time steps to generate its final score. This allows our approach to identify important features that have delayed impact on the predictive distribution, thus better capturing temporal dynamics in feature importance.

It is well known that the evaluation of explainability methods is challenging because no ground truth identification of important features exists for real world data (Doshi-Velez & Kim, 2017). A

common approach is to evaluate the impact of removing features that are highlighted as important by the explainability method (Lundberg et al., 2020) by masking them with a prior value. For time series data, an additional challenge for evaluation is that since observations of the same feature are related, removing the information from an important observation is non-trivial. This fact is under-explored in prior work. We present a detailed investigation of masking strategies to remove important observations and demonstrate their large impact on explainability performance evaluation. We present several masking strategies with complementary properties that we propose should be part of a comprehensive evaluation scheme.

In summary, our main contributions are:

- We propose a new time series feature importance method that accounts for the temporal dependence between different observations of the same feature and computes the impact of a feature observation to predictions over a window of time steps.
- We conduct a detailed investigation of evaluation approaches for time series model explainability and present several complementary masking strategies that we propose should be part of a robust evaluation scheme.
- We expand synthetic datasets from Tonekaboni et al. (2020) to explicitly evaluate the ability of explainability methods to capture shifted temporal dependencies.
- We conduct extensive experiments and demonstrate that our approach leads to a significant improvement in explanation performance on real-world data, and is more stable under different evaluation settings.

## 2 RELATED WORK

A wide range of saliency methods have been proposed in the literature. These include *gradient based methods* such as Integrated Gradients (Sundararajan et al., 2017), GradientSHAP (Erion et al., 2020), Deep-LIFT (Shrikumar et al., 2017), and DeepSHAP (Lundberg & Lee, 2017), which leverage gradients of model predictions with respect to input features to generate importance scores. *Feature removal based* methods such as feature occlusion (Zeiler & Fergus, 2014) and feature ablation (Suresh et al., 2017) are model-agnostic methods which remove a feature from the input data and measure the changes in model predictions. *Model based* saliency methods, such as Choi et al. (2016); Song et al. (2018); Xu et al. (2018); Kaji et al. (2019) for attention-based models, use the model architecture, in this case the attention layers, to generate importance scores. However, when applied to time series models, these explainability methods do not directly consider the temporal nature of the problem and have been shown to underperform (Ismail et al., 2020).

In contrast there has been little work on saliency methods for time series. TSR (Ismail et al., 2020) separates the importance calculation along the time and feature input dimensions to improve performance. FIT (Tonekaboni et al., 2020) measures each observation's importance to the prediction change at the same time step using a KL-divergence based score. Dynamask (Crabbé & van der Schaar, 2021) is a perturbation method that learns a mask of the input feature matrix to highlight important observations. We propose a new feature removal based saliency method that explicitly accounts for the temporal nature of the data in a novel way.

To understand how our work relates to other feature removal based methods we can utilize the unified framework presented in Covert et al. (2021) that categorizes feature removal based methods into three dimensions: feature removal method, model behaviour explained and summarization method. Approaches such as feature occlusion (Zeiler & Fergus, 2014), feature ablation (Suresh et al., 2017) and RISE (Petsiuk et al., 2019) replace the removed features with a specific baseline value. Like us, approaches such as FIT (Tonekaboni et al., 2020) and FIDO-CA (Chang et al., 2019) use a generator to replace the removed features. In terms of model behaviour explained, like most methods we aim to explain the model prediction before and after feature removal, while methods such as INVASE (Yoon et al., 2019) measure the change in the loss instead. The summarization method of an explainability approach describes how the importance score is generated. Feature occlusion uses the L1 norm of the difference in model predictions, which does not account for the temporal nature of the problem, while FIT uses the KL-divergence between the prediction with and without a feature at the *same* time step. Our summarization method differs from these approaches by accounting for the importance of a feature observation over *multiple* time steps, improving our ability to capture temporal patterns in feature importance.

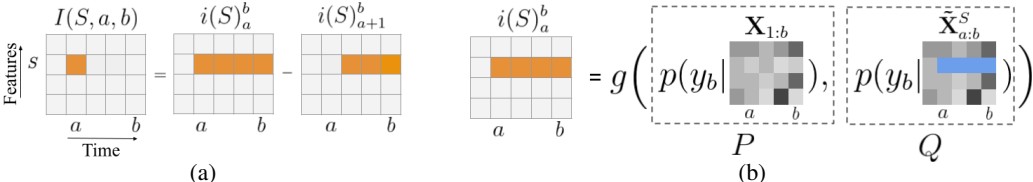

Figure 1: (a) The importance of a feature $S$ at a single time step $a$ on the prediction at time $b$ is given by the difference of the importance of $S$ over multiple time steps $a \ldots b$ and $a+1 \ldots b$, denoted by $i(S)_a^b$ and $i(S)_{a+1}^b$ respectively. (b) $i(S)_a^b$ is obtained by comparing $P$, the model prediction given the full input feature matrix to time $b$, $\mathbf{X}_{1:b}$, and $Q$, the prediction when the feature $S$ from $a \ldots b$ are unknown, indicated in blue and denoted by $\tilde{\mathbf{X}}_{a:b}^S$. These features are removed using a generative model. $g$ is a measure used to compare the distributions $P$ and $Q$.

## 3 OUR APPROACH: CAPTURING TEMPORAL DEPENDENCIES IN FEATURE IMPORTANCE

In this section we introduce our approach WinIT. We assume that a multivariate time series has been observed up to time step $T$ and a black-box model, $f_\theta$, has generated a predictive distribution at each time step. The change in predictions over time represents an updated state of the model as new data becomes available. WinIT assigns importance by summarizing how a given observation has impacted model predictions over time.

We denote $\mathbf{X} \in \mathbb{R}^{D \times T}$ as a sample of a multi-variate time series with $D$ features and $T$ time steps. We also let $\mathbf{x}_t := \mathbf{X}_{.,t} \in \mathbb{R}^D$ be the set of all feature observations at a particular time $1 \leq t \leq T$ and $\mathbf{X}_{1:t} := [\mathbf{x}_1; \mathbf{x}_2; \ldots; \mathbf{x}_t] \in \mathbb{R}^{D \times t}$. Let $y_t \in \{1 \ldots K\}$ be the label at each time step for a classification task with $K$ classes. The model, $f_\theta$, estimates the conditional distribution $p(y_t|\mathbf{X}_{1:t})$ at each time step $t$. Let $S \subseteq \{1, \ldots, D\}$ be a subset of features of interest, $S^c$ its complement, and $\mathbf{x}_t^S$ be the observations of that subset at time $t$.

A given observation $\mathbf{x}_t^S$ can impact predictions at all subsequent time steps. Our main goal is to generate an importance score for each observation that summarizes its impact on the following prediction time steps. We denote $I(S, a, b)$ as the importance of the feature set $S$ at time $t = a$ on the prediction at time $t = b$, where $b < a + N$ is bounded above by a window size $N$. To compute the total importance for $S$ at $a$, we aggregate $I(S, a, b)$ across the window $N$.

### 3.1 IMPORTANCE SCORE FORMULATION

To define $I(S, a, b)$ we first determine the importance $i(S)_a^b$ to the prediction at time $t = b$ of $S$ over *multiple* time steps $a \ldots b$, which we define in Eq. 2 using a feature removal based approach. The importance of $S$ at a *single* time step $t = a$ on the prediction at $t = b$ is then given by:

$$I(S, a, b) = \begin{cases} i(S)_a^b - i(S)_{a+1}^b, & a < b < a + N \\ i(S)_a^a & b = a \end{cases} \tag{1}$$

where $N$ is the window size. Since there is a dependence between subsequent observations of the same feature, eg. $\mathbf{x}_a^S$ and $\mathbf{x}_{a+1}^S$, in order to isolate the importance of feature set $S$ at $a$ on the prediction at $b$ we can not simply remove a single observation, and instead we must account for the importance of subsequent observations of $S$. This is done by formulating the importance in Eq. 1 as a difference between the importance over the time intervals $[a, b]$ and $[a + 1, b]$. This is represented pictorially in Figure 1a.

In our formulation, the importance $i(S)_a^b$ over multiple time steps in Eq. 1 can be obtained using a variety of feature removal based importance methods (Covert et al., 2021). Let $\mathbf{X}_{1:b}$ denote the input time series when all features from time steps $1 \ldots b$ are known and $\tilde{\mathbf{X}}_{a:b}^S := \mathbf{X}_{1:a-1}, \mathbf{X}_{a:b}^{S^c}$ denote the input features from $1 \ldots b$ when the feature set $S$ is unknown at $t \in [a, b]$. We focus on approaches that evaluate the impact of feature removal on the model prediction and consider the

following measures to compare the two distributions $P = p(y_b|\mathbf{X}_{1:b})$ and $Q = p(y_b|\tilde{\mathbf{X}}_{a:b}^S)$:

$$i_{\text{PD}}(S)_a^b = \|P - Q\|_1; \; i_{\text{KL}}(S)_a^b = D_{\text{KL}}(P\|Q); \; i_{\text{JS}}(S)_a^b = \frac{D_{\text{KL}}(P\|M) + D_{\text{KL}}(Q\|M)}{2}. \quad (2)$$

where $M = \frac{P+Q}{2}$. Here PD, KL and JS denote prediction difference, KL divergence and Jensen-Shannon divergence respectively. This is represented pictorially in Figure 1b, where $g$ denotes the measure used to compare the distributions $P$ and $Q$.

Prediction difference is similar to methods such as Feature Occlusion (Suresh et al., 2017), while KL based scores are used in Tonekaboni et al. (2020); Yoon et al. (2019), however these formulations are different to that in Eq. 2. Based on empirical evaluation, we adopt Prediction Difference for $i(S)_a^b$ in our approach and present details of the empirical comparison in Section 5.

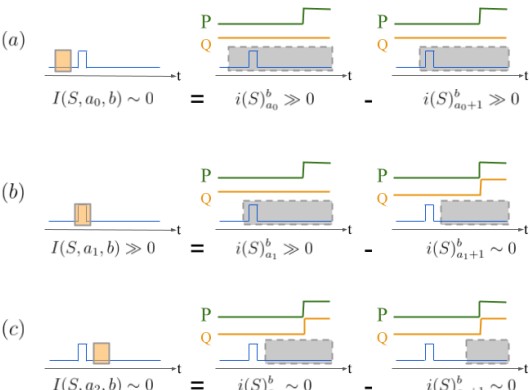

To gain further intuition into the formulation of our importance score, we consider different scenarios depicted in Figure 2. In our toy example, there is a signal in the feature set $S$ at time step $t = a_1$ that causes the prediction to change at a future time step $t = b$. For $t = a_0 < a_1$, both $i(S)_{a_0}^b$ and $i(S)_{a_0+1}^b$ would be large as the signal is contained in both intervals $[a_0, b]$ and $[a_0 + 1, b]$. For $t = a_2 > a_1$, both $i(S)_{a_2}^b$ and $i(S)_{a_2+1}^b$ would be small as the signal is not contained in either $[a_2, b]$ or

Figure 2: A toy example in which there is a signal in feature $S$ at time $t = a_1$ that causes a prediction change at time $t = b$, where $a_0 < a_1 < a_2 < b$. The importance scores for $S$ at $t = a_0, a_1, a_2$ are illustrated in (a), (b) and (c) respectively. $P$ indicates the prediction distribution with all inputs, while $Q$ is the prediction with the the grey feature observations removed.

$[a_2 + 1, b]$. Thus in both cases, the importance $I(S, a_0, b)$ and $I(S, a_2, b)$ would be small. In contrast, $I(S, a_1, b)$ would be large as $i(S)_{a_1}^b$ is large because the signal is in $[a_1, b]$ while $i(S)_{a_1+1}^b$ is small because the signal is not in $[a_1 + 1, b]$. Thus our formulation captures as important the moment of the feature change that would cause the predictions to change.

### 3.2 Feature Removal with Generative Model

To obtain the importance score for WinIT we need to compute $p(y_b|\tilde{\mathbf{X}}_{a:b}^S)$, where $\tilde{\mathbf{X}}_{a:b}^S := \mathbf{X}_{1:a-1}, \mathbf{X}_{a:b}^{S^c}$. This is the predictive distribution where all feature values are known up to time step $b$, except for feature set $S$ from $a$ to $b$. Approaches to feature removal include setting features to a default value (Petsiuk et al., 2019), sampling from a fixed distribution (Suresh et al., 2017), or removing features by replacing them with samples from a conditional generative model (Chang et al., 2019; Tonekaboni et al., 2020). We use a generative approach to replace the removed observations with plausible counterfactual alternatives.

The marginal probability we would like to compute can be written as:

$$p(y_b|\tilde{\mathbf{X}}_{a:b}^S) = p(y_b|\mathbf{X}_{1:a-1}, \mathbf{X}_{a:b}^{S^c}) = \mathbb{E}_{\hat{\mathbf{X}}_{a:b}^S \sim p(\mathbf{X}_{a:b}^S|\mathbf{X}_{1:a-1})}[p(y_b|\mathbf{X}_{1:a-1}, \mathbf{X}_{a:b}^{S^c}, \hat{\mathbf{X}}_{a:b}^S)] \quad (3)$$

where the conditional distribution is evaluated at specific values of $\mathbf{X}_{a:b}^S$ and averaged over the distribution of all values of $\mathbf{X}_{a:b}^S$. To estimate the distribution of $\mathbf{X}_{a:b}^S$, we use a non-deterministic feature generator $G_S$ that uses past observations, from time $1 \ldots a - 1$, to generate a distribution of features $S$ at time $a \ldots b$ for $a \le b < a + N$

$$p(\mathbf{X}_{a:b}^S|\mathbf{X}_{1:a-1}) = G_S(\mathbf{X}_{1:a-1}, n + 1). \quad (4)$$

Here the second argument in $G_S$ is the number of time steps the generator is going to generate, where $b = a + n$ and $0 \le n < N$. We sample $L$ times from the distribution in Eq. 4 and average to obtain the desired output in Eq. 3. Details are presented in Algorithm 1 in Appendix A.4.

### 3.3 Aggregation

Our method provides an importance score for a feature subset $S$ at time $t = a$ to the prediction at $t = b$. This allows the same feature observation to have varying importance for different prediction time steps, and provides a more granular view of temporal dependencies. However, it may also be of interest to provide a single importance score for each time step that summarizes the overall impact of $S$. In this case we aggregate the importance scores across the forward prediction time window:

$$\mathcal{I}(S, a) = \frac{1}{\hat{N} + 1} \sum_{b=a}^{a+\hat{N}} I(S, a, b), \quad \hat{N} = \min(N - 1, T - a) \tag{5}$$

In our framework we use $\mathcal{I}(S, a)$ to compute the importance for feature set $S$ at time point $a$, and use it to compare WinIT against leading baselines.

## 4 Explainability Evaluation in Time Series

In most real world applications, the true importance of features to a model's prediction are not known. A common evaluation strategy for feature importance based explainability methods is to remove important features and measure the difference in the model's performance (Hooker et al., 2019; Lundberg et al., 2020; Tonekaboni et al., 2020). Important features are assumed to be more informative to the model predictions, therefore the more important a removed observation is, the bigger the performance drop. For time-series, the important feature is *masked* by replacing it with its value at the previous time step or its mean value, for example. However, since subsequent observations of the same feature are correlated, we cannot remove the information of an important observation by only masking that single value. Tonekaboni et al. (2020) removes the information in an important observation, $\mathbf{x}_t^S$, by masking all the following observations of that feature in the time series, ie. $\mathbf{x}_{t'}^S \mapsto \hat{\mathbf{x}}_{t'}^S = \mathbf{x}_{t-1}^S \ \forall t' \geq t$. We refer to this ask the END masking strategy. However, when evaluating explainability methods by removing important features using END masking, the actual number of observations masked may be very different depending on where they are in the time series - and thus provide an unfair advantage methods that flag earlier time points as important. On the other hand, forcing the number of masked observations to be equal may also introduce bias as information spread over more time steps will be masked less. To this end, we adapt three masking strategies, defined below, to investigate their impact on results: (1) **END**: When an observation is declared important, we mask all the subsequent observations of that feature until the end; (2) **STD**: When an observation is declared important, we mask all the subsequent observations, until the feature value has changed more than $\alpha$ standard deviations with respect to the first masked observation. (3) **STD-BAL**: We designate a fixed number of masked observations, $n_m$. Here, we iterate over the most important feature-time $(S, t)$ instances (in order of importance) and apply STD until the number of masked observations reaches $n_m$.

The STD strategy stops masking once the observed value has changed significantly, reflecting the dynamics of the underlying time series. It is therefore more conservative than the END approach, as it will mask fewer observations. Furthermore, unlike the END approach, it does not favour explainability approaches that identify important observations earlier in the time series. STD-BAL follows a similar approach to STD, but fixes the number of observations masked. By comparing STD and STD-BAL, we can see the impact on evaluation from fixing the number of important observations to remove versus the number of observations that are masked. As we will show, the choice of masking strategy can lead to very different conclusions about the performance of competing explainability approaches. Given their complementary focus, we propose different masking strategies should be part of a robust evaluation scheme for time series explainability.

## 5 Experiments

For all experiments we use 1-layer GRU models for $f_\theta$ with 200 hidden units. We use Adam optimizer with learning rate $10^{-3}$ ($10^{-4}$ for MIMIC-III) and weight decay $10^{-3}$ to train the model. In practice, $S$ can be a single feature or a set selected by prior domain knowledge. For example, in the healthcare setting, $S$ can be a subset containing certain lab measurements, or indicators of different

Table 1: Performance on the simulated datasets. For WinIT we use a window size of 10. All evaluations are conducted over 5-fold cross-validation and averaged.

| METHOD | SPIKE | | DELAYED-SPIKE | | STATE | |
|---|---|---|---|---|---|---|
| | AUPRC | MEAN RANK | AUPRC | MEAN RANK | AUPRC | MEAN RANK |
| DEEP LIFT | $0.887_{\pm0.107}$ | $1.207_{\pm0.189}$ | $0.883_{\pm0.143}$ | $1.312_{\pm0.367}$ | $0.022_{\pm0.0}$ | $284.54_{\pm1.237}$ |
| GRADSHAP | $0.426_{\pm0.044}$ | $2.739_{\pm0.242}$ | $0.465_{\pm0.112}$ | $2.674_{\pm0.445}$ | $0.021_{\pm0.0}$ | $286.51_{\pm1.462}$ |
| IG | $0.378_{\pm0.065}$ | $3.0_{\pm0.21}$ | $0.421_{\pm0.102}$ | $3.081_{\pm0.505}$ | $0.022_{\pm0.0}$ | $285.12_{\pm1.188}$ |
| FO | $0.187_{\pm0.332}$ | $5.221_{\pm2.794}$ | $0.005_{\pm0.001}$ | $15.616_{\pm4.818}$ | $0.027_{\pm0.0}$ | $198.44_{\pm1.964}$ |
| AFO | $0.944_{\pm0.01}$ | $1.011_{\pm0.01}$ | $0.005_{\pm0.001}$ | $26.051_{\pm13.465}$ | $0.027_{\pm0.0}$ | $201.64_{\pm1.707}$ |
| FIT | $0.803_{\pm0.1}$ | $1.019_{\pm0.032}$ | $0.002_{\pm0.001}$ | $167.615_{\pm34.063}$ | $0.23_{\pm0.013}$ | $116.08_{\pm12.44}$ |
| DYNAMASK | $0.503_{\pm0.0}$ | $\mathbf{1.0}_{\pm0.0}$ | $0.502_{\pm0.002}$ | $1.125_{\pm0.171}$ | $\mathbf{0.278}_{\pm0.003}$ | $\mathbf{79.91}_{\pm0.72}$ |
| WINIT | $\mathbf{1.0}_{\pm0.0}$ | $\mathbf{1.0}_{\pm0.0}$ | $\mathbf{0.996}_{\pm0.009}$ | $1.004_{\pm0.008}$ | $0.26_{\pm0.01}$ | $84.69_{\pm2.058}$ |

chronic diseases. For simplicity, in all experiments we set $S = \{i\}$ to contain a single feature. We also use a 1-layer GRU model for the generator with 50 hidden units, and train it by fitting a Gaussian distribution with diagonal covariance to reconstruct each feature $N$ time steps forward. We use Adam optimizer with learning rate $10^{-4}$, weight delay $10^{-3}$ and 300 epochs with early stopping. To estimate the predicted distribution with masked features we average $L = 3$ Monte Carlo samples from the generator, as taking more samples does not improve performance significantly. We measure stability by splitting the training set into 5 folds and report results averaged across the folds with corresponding standard deviation error bars. To demonstrate that WinIT performs equally well on other common architectures we apply it to a 3-Layer GRU, LSTM, and ConvNet on MIMIC-III and results are shown in Appendix A.6. All the results with STD masking are with $\alpha = 1$. All experiments were performed with 40 Intel Xeon CPU@2.20GHz cores and Nvidia Titan V GPU.

We compare our approach against leading baselines: **Gradient-based methods**. This includes methods that use the gradient of the prediction with respect to the feature-time input. We compare against Deep-LIFT (Shrikumar et al., 2017), DeepSHAP (Lundberg & Lee, 2017) and Integrated Gradients (IG) (Sundararajan et al., 2017). For these comparisons we make use of the implementation provided in the Captum library (Kokhlikyan et al., 2020). **Feature occlusion** (FO) (Suresh et al., 2017) Feature importance is assigned based on the difference in prediction when each feature $i$ is replaced with a random sample from the uniform distribution. We also include Augmented Feature Occlusion (AFO) which was introduced in Tonekaboni et al. (2020) to avoid generating out-of-distribution samples. **Feature Importance in Time (FIT)** (Tonekaboni et al., 2020) assigns importance to an observation based on the KL-divergence between the predictive distribution and a counterfactual where the rest of the features are unobserved at the last time step. **Dynamask** (Crabbé & van der Schaar, 2021) is a perturbation-based method for time series data that learns a mask for the input sequence, that is almost binary.

## 5.1 SIMULATED DATASET

**Dataset** Spike is a benchmark experiment presented in Tonekaboni et al. (2020) which uses a multivariate dataset composed of 3 random NARMA time series with random spikes. The label is 0 until a spike occurs in the first feature, at which point it changes to 1 for the rest of the sample. We also add a new delayed version of this dataset, Delayed Spike, where the point at which the label changes to 1 is two time steps after the spike occurs. In this case, the observation that causes the label to change to 1 is the first spike two time steps before. We also include the synthetic benchmark dataset State (Tonekaboni et al., 2020), which uses a non-stationary Hidden Markov Model to generate observations over time and presents a more challenging task.

**Evaluation** Since here we know the exact reason for the positive label, we can use this reason as the ground truth importance to evaluate the explanations. The explanation label at the specific observations that cause the prediction label to change is 1, otherwise it is 0. We evaluate our feature importance using AUPRC. In addition, we introduce the mean rank metric – for each instance we compute the rank of the feature importance score of the positive ground truth observations within that instance. We take the average of this rank across all test instances. See Appendix A.5 for further discussion of the metrics. Mean rank is especially effective in assessing the performance of the spike and spike-delay datasets, as there is only one positive ground truth observation per instance.

Table 2: Performance on the MIMIC-III mortality task using END masking.

| | PERFORMANCE (TOP 5%) | | PERFORMANCE (K=50) | |
| | AUC DROP | PRED. CHANGE | AUC DROP | PRED. CHANGE |
| --- | --- | --- | --- | --- |
| DEEP LIFT | $0.025_{\pm 0.002}$ | $0.031_{\pm 0.002}$ | $0.021_{\pm 0.003}$ | $0.029_{\pm 0.002}$ |
| GRADSHAP | $0.022_{\pm 0.004}$ | $0.03_{\pm 0.002}$ | $0.021_{\pm 0.004}$ | $0.027_{\pm 0.002}$ |
| IG | $0.021_{\pm 0.003}$ | $0.03_{\pm 0.002}$ | $0.02_{\pm 0.004}$ | $0.027_{\pm 0.002}$ |
| FO | $0.047_{\pm 0.004}$ | $0.045_{\pm 0.003}$ | $0.067_{\pm 0.008}$ | $0.057_{\pm 0.004}$ |
| AFO | $0.049_{\pm 0.009}$ | $0.05_{\pm 0.002}$ | $0.066_{\pm 0.007}$ | $0.06_{\pm 0.003}$ |
| FIT | $0.066_{\pm 0.007}$ | $0.056_{\pm 0.003}$ | $0.073_{\pm 0.007}$ | $0.07_{\pm 0.003}$ |
| DYNAMASK | $0.068_{\pm 0.004}$ | $0.068_{\pm 0.004}$ | $0.082_{\pm 0.007}$ | $0.076_{\pm 0.004}$ |
| WINIT | $\mathbf{0.098}_{\pm 0.008}$ | $\mathbf{0.08}_{\pm 0.004}$ | $\mathbf{0.099}_{\pm 0.008}$ | $\mathbf{0.079}_{\pm 0.004}$ |

**Results.** The results on simulated data are presented in Table 1. We see that WinIT outperforms other methods on the spike and delayed spike datasets in AUPRC and is on par with Dynamask in mean rank. Unlike WinIT, the performance of other feature removal based meth-

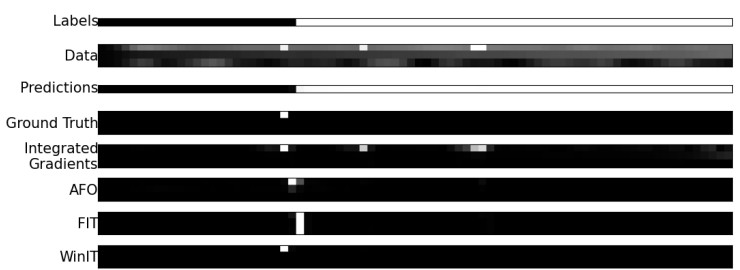

Figure 3: Saliency map for delayed-spike dataset.

ods, FO, AFO and FIT, degrades significantly for the delayed spike experiment compared to spike. This indicates that the feature removal based importance formulation in WinIT is more robust to delayed importance signals. Similarly we note that Dynamask and the gradient based approaches are able to adapt to the delayed signal. The strong performance on the state dataset also shows WinIT's robustness to non-stationary signals. Saliency maps for the delayed spike experiment are shown in Figure 3. WinIT captures the important feature in this setting, while gradient-based method, IG, also highlights subsequent spikes. A more complete set of saliency maps are shown in Appendix A.2.

Note, unlike other methods Dynamask encourages the importance score to be almost binary. As a result it generates a substantial number of repeated values and this impacts its evaluation on simulated data. The results in Table 1 use the trapezoid rule to compute AUPRC, following Crabbé & van der Schaar (2021). Without interpolation (Davis & Goadrich, 2006), Dynamask's AUPRC is much lower ($0.038 \pm 0.0$ for the state experiment), while other methods AUPRC values remain similar. Similar problem arises for mean rank. See Appendix A.5.2 for further analysis.

## 5.2 MIMIC-III MORTALITY

**Dataset** MIMIC-III is a multivariate clinical time series dataset with a range of vital and lab measurements taken over time for around 40,000 patients at the Beth Israel Deaconess Medical Center in Boston, MA (Johnson et al., 2016). It is widely used in healthcare and medical AI-related research. There are multiple tasks associated, including mortality, length-of-stay prediction, and phenotyping (Wang et al., 2020). We follow the pre-processing procedure described in Tonekaboni et al. (2020) and use 8 vitals and 20 lab measurements hourly over a 48-hour period to predict patient mortality.

**Evaluation** We report performance using both global and local evaluation, as is done in Tonekaboni et al. (2020). The global method corresponds to measuring the impact of removing the observations with the highest (top 5%) importance scores across the test set. While the local method corresponds to removing the observations with the highest (top $K = 50$) importance scores within each instance. For both experiments we present the metrics AUC drop and average prediction change when important features are masked. Average prediction change, defined as the average absolute prediction change, allows us to evaluate the explainability method without relying on the ground truth labels in the dataset. Our base model has an AUC of $0.81 \pm 0.003$ in the test set. For the masking method, we use the END masking method from Tonekaboni et al. (2020) for comparison purposes (See Section 4). We explore the impact of other masking strategies in Section 5.3.

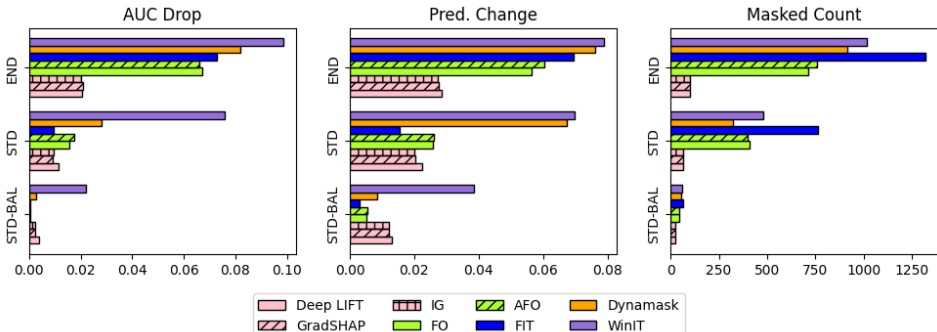

Figure 4: AUC Drop, Mean Prediction Change and the number of masked observations for STD-BAL, STD and END masking method for different explainability methods on $K = 50$.

**Results.** Our main results are presented in Table 2. WinIT uses a window size of 10. We can see that WinIT significantly outperforms all the other methods for both the global top 5% and local $K = 50$ experiments. WinIT improves over baselines both in terms of identifying observations that impact AUC most and the change in prediction. WinIT has a 44% increase in AUC Drop and 15% increase in prediction change over the leading baseline Dynamask on the top 5% experiment. Compared to other feature removal based methods, FO, AFO and FIT, the gain is even larger.

## 5.3 ADDITIONAL ANALYSIS

**Impact of masking strategy.** As discussed in Section 4, when evaluating explainability methods by removing features and measuring the impact on performance, the strategy used to mask observations can introduce bias into the evaluation. In Figure 4, we show results for the $K = 50$ MIMIC-III mortality task using the masking strategies outlined in Section 4. We see that masking has a large impact on the performance. WinIT outperforms other methods across all masking strategies, however the relative performance of gradient versus feature removal based methods, FO, AFO and FIT, changes between the END and STD/STD-BAL methods. Although for a given explainability method the AUC drop and prediction change is highest with the END strategy, which tends to mask the most observations, this approach penalizes approaches (such as gradient-based methods) that have a bias towards more recent observations. This can be seen in Figure 5, which shows the counts of masked observations at each time step for different explainability methods[1]. We see that the gradient-based method IG masks more recent time steps for all masking strategies, likely due to the vanishing gradients phenomenon (Pascanu et al., 2012). In contrast the masked observations for methods such as AFO, WinIT and Dynamask are more evenly distributed across time steps.

STD masking reduces the evaluation bias of END masking against explanation methods that flag features later in the time series. While STD-BAL makes sure the total number of observation masked are roughly the same across all methods. Given their impact on evaluation, these masking strategies should be used as part of a broader set of evaluation methods in order to make the evaluation of time series explainability methods more robust.

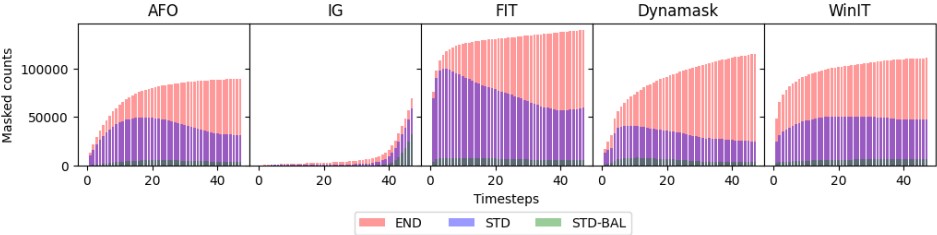

Figure 5: Count of masked observations for Integrated Gradients, FIT, Dynamask and WinIT at each time step in the MIMIC-III $K = 50$ performance drop experiment. Note that the bar charts are overlapping, instead of stacked.

---

[1]Note the distribution of masked observations for Gradient SHAP and DeepLIFT is similar to IG; and the distributions of FO is similar to AFO.

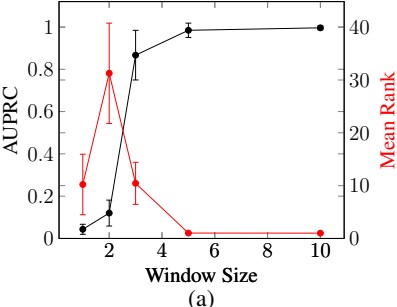 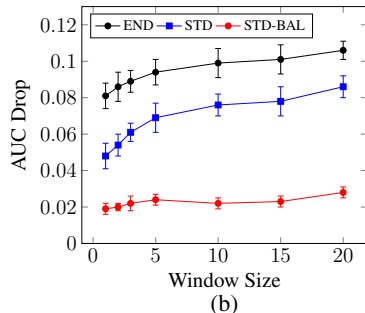

(a)                                    (b)

Figure 6: (a) WinIT performance on the Delayed Spike dataset as the window size from 1 to 10. (b) WinIT AUC Drop performance on the MIMIC III mortality task (K=50) as the window size from 1 to 20 for different masking schemes. Error bars show the standard deviation across the folds.

**Effect of window size** $N$**.**   WinIT assigns importance to observations by aggregating the impact on predictions over a window of subsequent time steps. As the window size increases, WinIT captures longer range interactions between important signals and changes in predictions, which can lead to better performance since some signals can be heavily delayed. In Figure 6a we analyze this effect for different window sizes on the delayed spike dataset, where the delay is exactly two time steps. As expected we see that as $N$ increases to 3, the performance, substantially improves. Note that $N = 3$ corresponds to looking backward two time steps (plus the current one). We also see that the performance quickly stabilizes after $N = 3$.

A similar pattern can be observed for the MIMIC-III mortality task shown in Figure 6b. We see that the AUC Drop increases with window size and then stabilizes, with a large range of choices of the hyper-parameter $N$ leading to similar results. We also note that there is a linear relationship between run time and window size, as shown in Appendix A.4, allowing for a reasonable hyper-parameter sweep to find a good choice of window size. This pattern holds for different masking strategies, demonstrating WinIT's robustness to evaluation strategy.

**Effects of Different** $i(S)_a^b$ **.**   We investigate the impact on performance of using different definitions of the importance to the prediction at $t = b$ of $S$ over multiple time steps $[a, b]$, $i(S)_a^b$, defined in Eq. 2. In Table 3, we show the AUC drop on the MIMIC-III mortality task using STD masking. We see that WinIT-PD gives some gain over the other approaches when dropping the top

Table 3: AUC Drop on MIMIC-III with STD masking using different $i(S)_a^b$ in WinIT.

|  | **Top 5%** | **K=50** |
| --- | --- | --- |
| WINIT-KL | $0.076_{\pm 0.006}$ | $0.072_{\pm 0.007}$ |
| WINIT-JS | $0.075_{\pm 0.007}$ | $0.072_{\pm 0.007}$ |
| WINIT-PD | $\mathbf{0.078}_{\pm 0.008}$ | $\mathbf{0.076}_{\pm 0.006}$ |

50 observations for each time series. We opt to use the prediction difference in our formulation. It is worth noting that all three approaches outperform the other baselines.

## 6   CONCLUSION

In this paper we propose WinIT, a feature removal based explainability method that produces a feature importance score for each observation in a multivariate time series. Unlike prior approaches, WinIT calculates the importance as the difference of the importance over multiple time steps and aggregates the importance of each observation over a window of predictions. Empirically, WinIT outperforms leading baselines on both synthetic and real-world datasets with more consistent performance under different evaluation strategies. For future work, we would like to tackle two possible limitations in WinIT. First, we would like to explore approaches to automatically determine the window size parameter introduced in our method. Secondly we would like to extend to time series explainability approaches that do not take the form of saliency maps, with a single importance for each observation, but which instead focus on identifying important higher-level patterns or trends over a series of observations, as we believe this form of output would further aid interpretability.

## REPRODUCIBILITY STATEMENT

To ensure the reproducibility of our work, we describe the implementation details of our approach and conduct evaluation experiments on publicly available datasets. In addition, we follow the procedures outlined in previous studies to process the data. The code for our work is publicly available at `https://github.com/layer6ai-labs/WinIT`, which includes the detailed settings for experiments.

## ACKNOWLEDGMENTS

We would like to thank the anonymous reviewers for their feedback, which helped improve our paper. We would also like to thank Gabriel Loaiza-Ganem for useful discussions. This work was conducted while Clayton Rooke and Jonathan Smith were at Layer 6 AI.

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

# A APPENDIX

## A.1 SUMMARY OF NOTATION

A summary of the notation used throughout the paper is provided in Table 4.

Table 4: Summary of Notation

| NOTATION | DESCRIPTION |
|---|---|
| $T$ | Number of time steps in the time series. |
| $D$ | Number of features in the time series. |
| $f_\theta$ | A trained model that outputs the predictive distribution given the input time series. |
| $\mathbf{X}$ | The time series of $D$ features and $T$ time steps. |
| $\mathbf{x}_t$ | The vector of all feature observations at time step $t$, $1 \leq t \leq T$. |
| $\mathbf{X}_{1:t}$ | The set of all observations in the time series $\mathbf{X}$ up to time $t$. |
| $K$ | The number of classes for a given classification task. |
| $y_t$ | The label at time step $t$ for a classification task with $K$ classes. |
| $S$ | The subset of features of interest, $S \subseteq \{1, \ldots, D\}$. |
| $S^c$ | The complement set to the features of interest $\{1, \ldots, D\} \setminus S$ |
| $\mathbf{x}_a^S$ | The observation of feature set $S$ at time-step $t = a$. |
| $\mathbf{X}_{a:b}^S$ | The observations of feature set $S$ from $t = a \ldots b$. |
| $\tilde{\mathbf{X}}_{a:b}^S$ | $\mathbf{X}_{1:b}$ with observations in feature set $S$ removed from $t = a \ldots b$. |
| $i(S)_a^b$ | The importance to the prediction at time $t = b$ of $S$ over time steps $t = a \ldots b$. |
| $I(S, a, b)$ | The importance of the feature set $S$ at time $t = a$ on the prediction at time $t = b$. |
| $G_S$ | A non-deterministic generator for feature $S$ using past observations. |
| $N$ | The maximum window size. |
| $\hat{N}$ | $\min(N - 1, T - a)$. |
| $\mathcal{I}(S, a)$ | The feature importance score aggregated over the window $b = a \ldots \hat{N}$. |

## A.2 SALIENCY MAPS

Saliency maps are one way of visualizing feature importance and have been found to be helpful for interpreting predictions for image classification in user studies Alqaraawi et al. (2020). In the multivariate time series setting Shen et al. (2020) found visualizations can also improve interpretation.

Saliency maps for the simulated spike and delayed spike datasets are shown in Figure 7 and 8 respectively. These maps illustrate that WinIT captures the most important features in both the spike and delayed spike setting. The gradient-based methods, capture the important observation, but also highlight subsequent spikes that do not impact the change in label.

We also shows the saliency maps for the MIMIC-III mortality experiments in Figure 9. We see that gradient-based methods tend to highlight later observations in the time series as important. In contrast FO and AFO tend to flag earlier points in the time series. FIT tends to mark many features within the same time step important, while the salient observations are more distributed in time for WinIT and Dynamask.

## A.3 AGGREGATION METHODS FOR WINIT

In section 3.3, we introduce a method of aggregation to compute the importance of feature set $S$ at timestep $t = a$ by taking the mean of the importance scores across a time window, according to Eq. 5. Here we consider an an alternate choice of aggregation, given below:

$$\mathcal{I}_{max}(S, a) = \max_{b \in [a, a + \hat{N}]} I(S, a, b) \tag{6}$$

where $\hat{N} = \max(N - 1, T - a)$. Table 5 shows the results for STD masking on the MIMIC-III mortality task with different aggregation choices. We see that while the performance for different aggregation methods are similar, taking the mean gives the best results, in terms of both AUC Drop and prediction change. One explanation for this is that if an observation is important for *multiple* future predictions, it is more important than the case when an observation is important for a single future prediction.

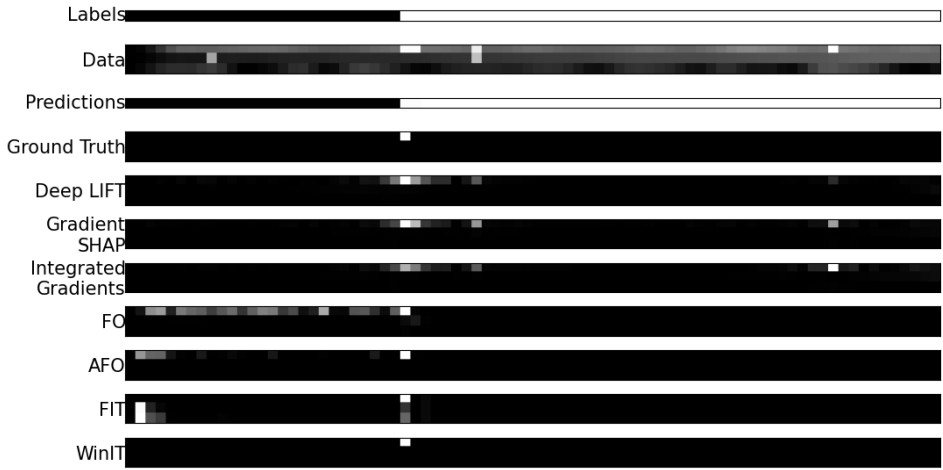

Figure 7: Saliency maps for simulation spike

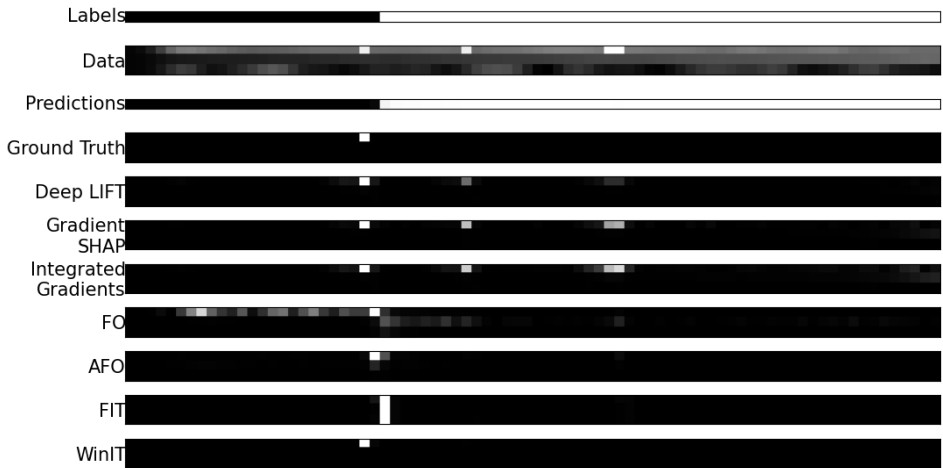

Figure 8: Saliency maps for delayed simulation spike

## A.4 ALGORITHM AND RUN TIME ANALYSIS

We provide an outline of the WinIT algorithm in Algorithm 1.

We also explore the run time for different window sizes, $N$, and present our results in Figure 10. We see that the run time has a linear relationship with the window size. Since the importance calculation for each feature is independent in our approach, we are able to parallelize this computation which improves the run time.

## A.5 EVALUATION OF SIMULATED DATA

### A.5.1 METRICS FOR SPIKE AND DELAYED SPIKE DATASETS

In this section we provide insight into the metrics used to evaluate the simulated spike and delayed spike datasets. These datasets have only one positively labelled observation for each time series and the ground truth explanation is known. Here we consider AUPRC and mean rank, which we use for evaluation and AUROC, another common metric, which we show is less suitable for this case.

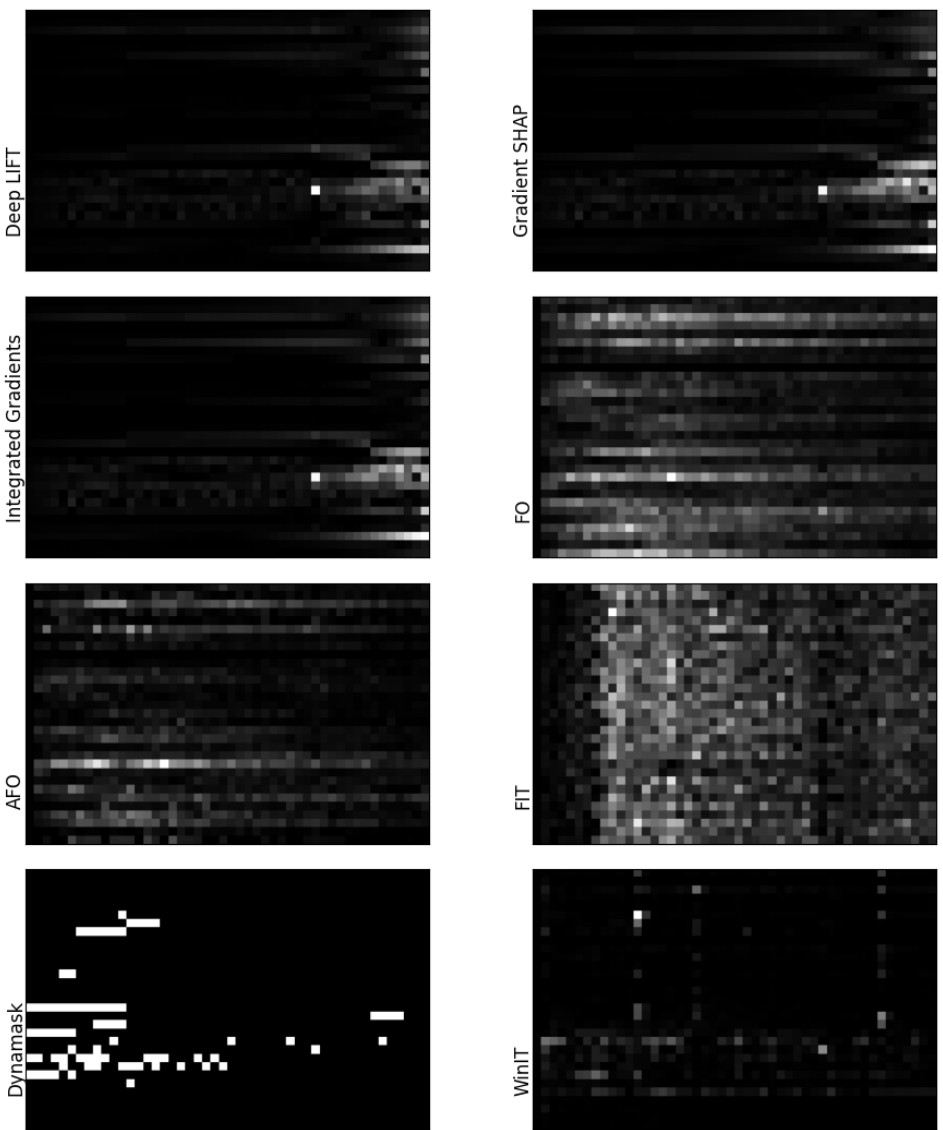

Figure 9: Saliency maps for MIMIC-III

Table 5: AUC Drop and Prediction Changes for different aggregation methods in WinIT on the MIMIC-III mortality task

|  | PERFORMANCE (TOP 5%) | | PERFORMANCE (K=50) | |
| --- | --- | --- | --- | --- |
|  | AUC DROP | PRED. CHANGE | AUC DROP | PRED. CHANGE |
| MEAN | $\textbf{0.078}_{\pm 0.008}$ | $\textbf{0.07}_{\pm 0.004}$ | $\textbf{0.076}_{\pm 0.006}$ | $\textbf{0.07}_{\pm 0.004}$ |
| MAX | $0.066_{\pm 0.008}$ | $0.061_{\pm 0.003}$ | $0.063_{\pm 0.004}$ | $0.059_{\pm 0.003}$ |

For simplicity, we assume all the feature importance values are distinct. Suppose there are $n$ observations in a time series. Let $1 \le r \le n$ be the rank of the feature importance among all observations in the time series.

AUROC can be interpreted as the probability that the prediction of the positively-labelled observation is greater than that of a negatively labelled observation. As there is 1 positively-labelled observation of rank $r$ in our case, this is the case for $n - r$ out of a total of $n - 1$ comparisons. Thus the AUROC is $\frac{n-r}{n-1}$.

---

**Algorithm 1 WinIT**

**Input:** $f_\theta$: trained Black-box predictor model; $S$: a subset of features of interest; $N$: feature importance window size; $G_S$: trained generative model; $\mathbf{X}_{1:T} \in \mathbb{R}^{D \times T}$: time series where $T$ is the max time and $D$ is the number of features; $L$: number of samples
**Output:** Importance score matrix $\mathcal{I} \in \mathbb{R}^T$

---

1: Train $G_S$ using $\mathbf{X}_{1:T}$ for all $0 \le n < N$
2: **for all** $t := T \dots 1$ **do**
3:    $p(y_t|\mathbf{X}_{1:t}) = f_\theta(\mathbf{X}_{1:t})$
4:    Initialize $i(S)_a^b := 0$ for all $a, b$.
5:    **for all** $n := 0 \dots \min(N-1, T-t)$ **do**
6:       $p(\mathbf{X}_{S,t:t+n}|\mathbf{X}_{1:t-1}) \approx G_S(\mathbf{X}_{1:t-1}, n+1)$
7:       **for all** $l := 1 \dots L$ **do**
8:          Sample $\hat{\mathbf{X}}_{S,t:t+n}^{(l)} \sim p(\mathbf{X}_{S,t:t+n}|\mathbf{X}_{1:t-1})$
9:          $p(\hat{y}_{t+n}^{(l)}) = f_\theta(\mathbf{X}_{1:t-1}, \mathbf{X}_{S^c,t:t+n}, \hat{\mathbf{X}}_{S,t:t+n}^{(l)})$
10:       **end for**
11:       $p(y_{t+n}|\mathbf{X}_{1:t-1}, \mathbf{X}_{S^c,t:t+n}) \approx \frac{1}{L}\sum_{l=1}^{L} p(\hat{y}_{t+n}^{(l)})$
12:       $i(S)_t^{t+n} = \|p(y_{t+n}|\mathbf{X}_{1:t+n}) - p(y_{t+n}|\mathbf{X}_{1:t-1}, \mathbf{X}_{S^c,t:t+n})\|_1$
13:       $I(S, t, n) = i(S)_t^{t+n} - i(S)_{t+1}^{t+n}$
14:    **end for**
15: **end for**
16: $\mathcal{I}(S, t) = \frac{1}{\min(N-1,T-t)+1} \sum_{n=0}^{\min(N-1,T-t)} I(S, t, n)$
17: Return $\mathcal{I}$

---

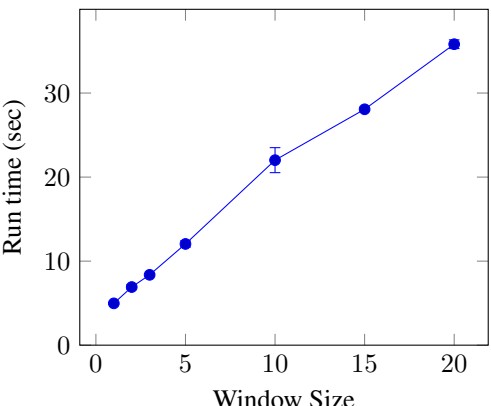

Figure 10: WinIT run time on the simulated spike delay task for different window sizes.

For AUPRC, as there is only 1 positively labelled observation, the recall can only be 1 or 0. When the recall is 1, the best precision is $1/r$ as the positively-labelled observation is of rank $r$. When the recall is 0, if the rank $r \neq 1$, the precision is also 0. Thus, for $r > 1$, there are only 2 points in the precision-recall curve, $(0,0)$ and $(1, \frac{1}{r})$. This means AUPRC is $\frac{1}{2r}$ for $r > 1$. For $r = 1$, AUPRC=1 by definition.

To summarize, when rank is $r$, AUROC=$\frac{n-r}{n-1}$ and AUPRC=$\frac{1}{2r}$. Since AUROC depends on $n$, which AUPRC only depends on $r$, AUROC is less sensitive to changes in performance than AUPRC. For example, our simulated spike dataset contains 3 features and 80 time points, therefore the number of observations is $n = 240$. If the positively-labelled feature is the 4th most important feature, i.e. $r = 4$, the AUROC is 0.9874 while the AUPRC is 0.125. If the rank is 10, AUROC is 0.958 and AUPRC is 0.05. This shows that AUPRC is much more sensitive to changes in rank that AUROC.

To compare performance across explainability methods on the spike and delayed spike datasets, we argue that mean rank and AUPRC provide the clearest comparison.

Table 6: AUPRC-interpolation, average precision, mean rank and mean average rank for all methods in State dataset.

|  | AUPRC-INTERPOLATION | AVERAGE PRECISION | MEAN RANK | MEAN AVG RANK |
|---|---|---|---|---|
| DEEP LIFT | $0.022_{\pm 0.0}$ | $0.022_{\pm 0.0}$ | $284.54_{\pm 1.237}$ | $289.15_{\pm 1.814}$ |
| GRADSHAP | $0.021_{\pm 0.0}$ | $0.021_{\pm 0.0}$ | $286.51_{\pm 1.462}$ | $290.4_{\pm 1.743}$ |
| IG | $0.022_{\pm 0.0}$ | $0.022_{\pm 0.0}$ | $285.12_{\pm 1.188}$ | $289.04_{\pm 1.939}$ |
| FO | $0.027_{\pm 0.0}$ | $0.027_{\pm 0.0}$ | $198.44_{\pm 1.964}$ | $198.44_{\pm 1.964}$ |
| AFO | $0.027_{\pm 0.0}$ | $0.028_{\pm 0.0}$ | $201.64_{\pm 1.707}$ | $201.64_{\pm 1.707}$ |
| FIT | $0.23_{\pm 0.013}$ | $0.194_{\pm 0.013}$ | $116.08_{\pm 12.442}$ | $116.37_{\pm 12.417}$ |
| DYNAMASK | $\mathbf{0.278}_{\pm 0.003}$ | $0.038_{\pm 0.0}$ | $\mathbf{79.909}_{\pm 0.72}$ | $179.77_{\pm 0.849}$ |
| WINIT | $0.26_{\pm 0.01}$ | $\mathbf{0.26}_{\pm 0.01}$ | $84.692_{\pm 2.058}$ | $\mathbf{84.692}_{\pm 2.058}$ |

### A.5.2 EVALUATING DYNAMASK ON SIMULATED DATASETS

As described in Section 5, Dynamask (Crabbé & van der Schaar, 2021) is expected to produce almost binary masks, while the importance scores for other methods are expected to be continuous values. This makes the evaluation of Dynmask on simulated data very sensitive to the way in which ties are resolved.

For completeness we present the impact of these choices on the AUPRC and mean rank metrics for the State dataset in Table 6. In particular we present mean rank where ties are resolved by taking the minimum rank and the average rank, where we refer to the latter as mean average rank. We also present the results of AUPRC computed with and without interpolation. We refer to AUPRC without interpolation as Average Precision. In the main text we use AUPRC with interpolation following the presentation in Crabbé & van der Schaar (2021), and note that elsewhere in the literature (Davis & Goadrich, 2006) it is argued that average precision is preferred. We see that Dynamask performance drops significantly when interpolation is not used in computing AUPRC and average vs minimum rank is used to resolve ties. Figure 11 shows the precision-recall curve and the two different types of interpolations.

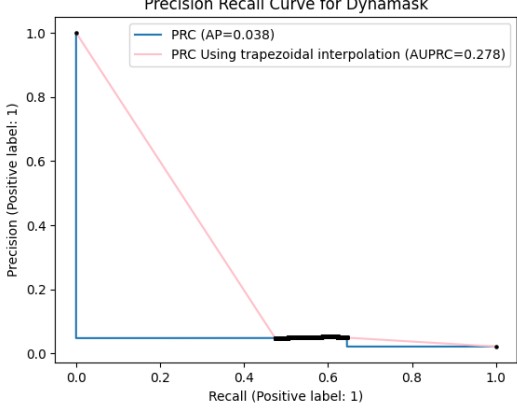

Figure 11: Precision Recall Curve for Dynamask. The black dots correspond to the precision and recall for different thresholds.

### A.6 ALTERNATIVE MODEL ARCHITECTURES FOR MIMIC-III MORTALITY

In this section we show that WinIT continues to perform well on different model architectures. We show the performance on the MIMIC-III Mortality dataset on three additional architectures: 3-Layer GRU, LSTM and ConvNet. We show the results using different masking scheme and show

the performance for the $K = 50$ task. The performance of explainability methods follows a similar trend for different model architectures.

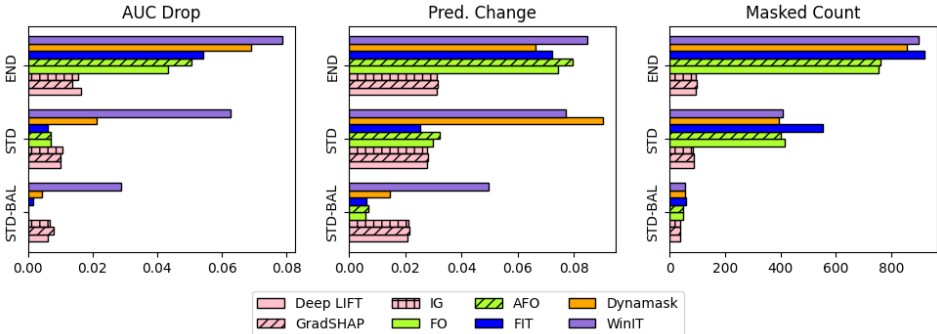

Figure 12: MIMIC-III Mortality performance with a ConvNet model architecture. (Base Model AUC=$0.746_{\pm 0.012}$)

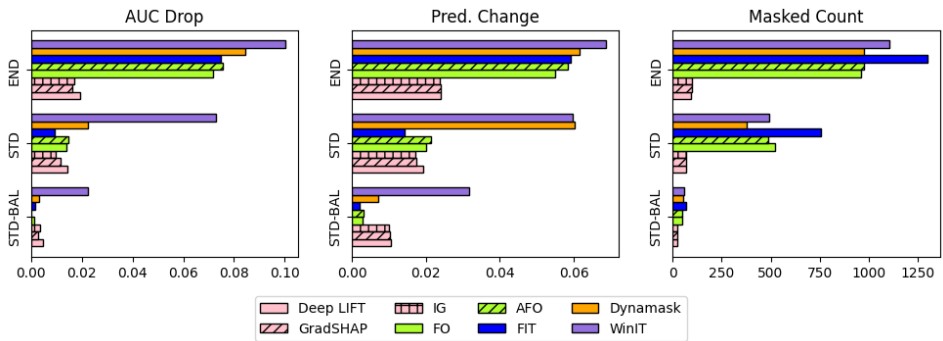

Figure 13: MIMIC-III Mortality performance with an LSTM model architecture. (Base Model AUC=$0.797_{\pm 0.003}$)

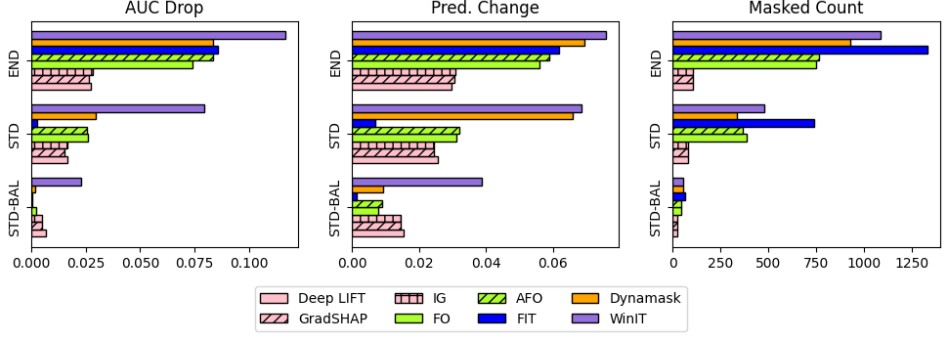

Figure 14: MIMIC-III Mortality performance with a 3-layer GRU model architecture. (Base Model AUC=$0.793_{\pm 0.003}$)

## A.7 TRAINING CURVES FOR GENERATORS

The training curves for some selected generators are shown in Figure 15. We apply early stopping to avoid overfitting.

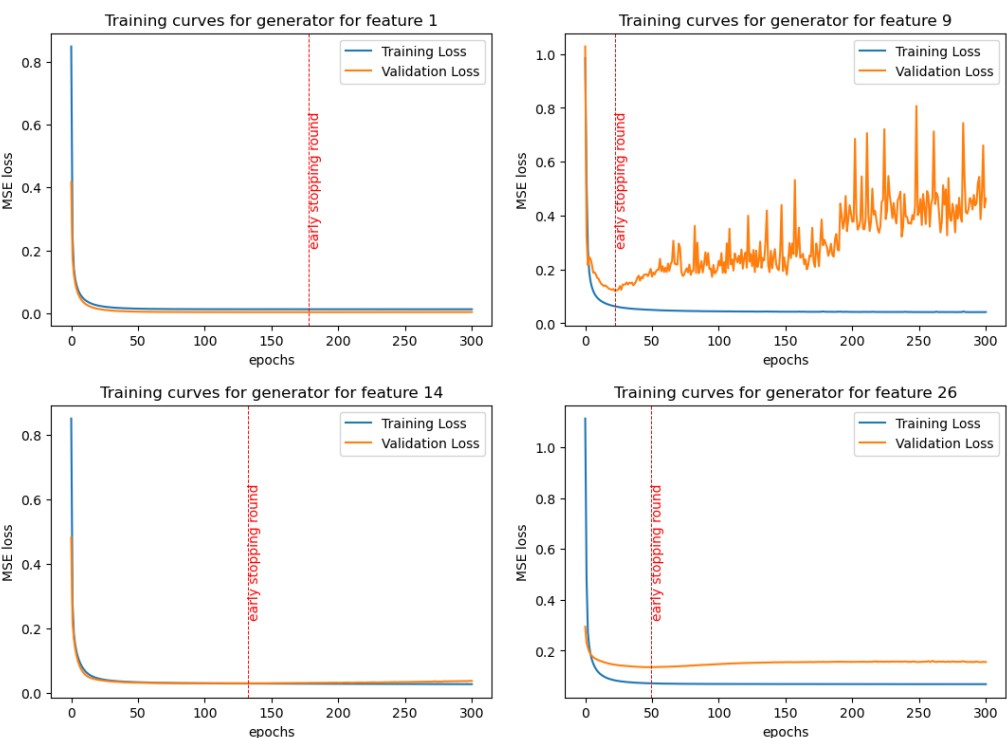

Figure 15: Example training curves for the generators

