# OpenReview forum: "Temporal Dependencies in Feature Importance for Time Series Prediction"
_ICLR.cc/2023/Conference — ICLR 2023 poster_

### Official Review · Reviewer_CWz4 · 2022-10-23

**Confidence:** 3
**Correctness:** 3
**Technical Novelty And Significance:** 3
**Empirical Novelty And Significance:** 2
**Recommendation:** 6

**Clarity, Quality, Novelty And Reproducibility:**

Clarity: Good. The paper is well organised and easy to follow. However, there need more justifications for the proposed measure.
Quality: I have not spotted a flaw.
Novelty: The idea is incremental.
Reproducibility: Most details are provided for reproducibility.



**Strength And Weaknesses:**

S1: The paper proposes a method for solving the challenges in explaining time series predictions.

S2: The paper is well structured and easy to follow.

S3: Experiments are comprehensive.


W1: The proposed method looks incremental. The measure of feature importance at a time step follows the measure of Tonekaboni et al. (2020) by extending it to prediction difference and Jensen Shannon divergence.

W2: The algorithm needs more justifications. The reason for the average over a window of size N is unclear to me. The window size N introduces another parameter to tune. Also, the definition of $i(S)^b_a$ needs more justifications. “We formulate the importance in Eq. 1 as a difference due to the time series nature of the data” is not clear enough to explain the intuition. Figure 1b has not been explained in the text. What does exactly $i(S)^b_a$ mean? Is a large (or small) value of $i(S)^b_a$ preferred? Without these explanations, Figure 2 is not easy to understand.  Furthermore, the predictions of time series with the information being masked should be presented. It is confusing that the predictions in all three cases (a) (b) and (c) are the same (the right hand sides before and after the minus symbol '-'). I wonder why $i(S)^b_a$s are different in the three cases when the predictions before and after masking are the same.

W3: The example of Figure 1 is too simple to show teh usefulness of the proposed method.  In Figure 1,  Y is determined by a single signal. Many time series predictions are determined by patterns (multiple signals). There needs an example to show that the proposed importance measure can handle pattern signals to convince readers that EinIt is useful for real world applications.

W4: In Figure 4, the AUC drops and Prediction changes are quite inconsistent. Can authors provide some explanations?


**Summary Of The Paper:**

The authors propose a Windowed Feature Importance in Time
(WinIT) method for explaining time series predictions. They have also investigated the evaluation approaches with different complementary masking strategies. Experiments have been conducted to show the proposed approach performs well in synthetics and real-world data sets.


**Summary Of The Review:**

It is a technically solid paper. The clarity of the proposed measure can be further improved. Without the explanations, it is difficult to judge the empirical impact of the methods for dealing with the pattern signals.

---

> ### Author Response · Authors · 2022-11-16
> **Response to Reviewer CWz4 (1/2)**
>
> We thank the reviewer for the detailed constructive feedback on our work. We appreciate the recognition of the comprehensive experimental study and that the presentation is easy to follow. Below we address the points raised by the reviewer in order and have updated our manuscript accordingly:
>
> 1.  We believe our approach significantly differs from the FIT method proposed in Tonekaboni et al. (2020), both in an important conceptual way, as well as in the details of the formulation, beyond the choice of measure used to compare changes in model prediction distributions with feature removal.
>
>     The key conceptual difference is that FIT defines the importance of a feature observation, $\mathbf{x}^S_{t=a}$, by measuring the change in the prediction with and without the feature at the *same* timestep. This effectively captures the case when the important observation and the prediction change occur at the same time step (ie. are instantaneous). This is analogous to our approach at $N=1$. In contrast, a key contribution of our method is to be able to identify important features when there is a delay between an important feature shift and a change in outcome. It is important for explanation methods for time series predictions to be able to perform well given such temporal dependencies. The performance of feature removal baselines FO, AFO, and FIT on the spike vs delayed-spike experiments, illustrates this limitation which our method overcomes.
>
>      In terms of the formulation details, we can express the FIT importance score as:
>      $\mathcal{I}(S,a) = D_{\text{KL}}(P||Q_1) - D_{\text{KL}}(P||Q_2)$,
>
>      where $P=p(y_a|\mathbf{X}_{1:a})$,      $Q_1=p(y_a|\mathbf{X} _ {1:a-1})$, and $Q_2 =p(y_a|\mathbf{X} _ {1:a-1},\mathbf{x}^S_a)$.
>
>      Here, the first term is an overall temporal distribution shift and the second term is the unexplained distribution shift when only $S$ at $t=a$ is observed, as described by the authors in Tonekaboni et al. (2020).
>
>      We can compare this to our score restricted to the case *without* windowing, at $N=1$:
>
>      $\mathcal{I}(S,a) = I(S,a,a) = g(P,Q)$
>      where $P$ is the same as above, $Q=p(y_a| \mathbf{X}_{1:a-1},\mathbf{x}^{S^c}_a)$,
>
>      and $g$ compares the distributions using either the prediction difference, KL or Jensen-Shannon divergence. Regardless of the choice of $g$, our score does not rely on the overall temporal distribution shift, and compares $P$ with the prediction when $S$ is unobserved, ie. $Q$. In contrast to the case of FIT, where they compare $P$ with both $Q_1$ and $Q_2$, our method is different, even for $N=1$. This can also be seen by the fact that our score at $N=1$ is always positive, while the importance in FIT can be negative. The extension of this formulation to the case $N>1$ (see Eqs.1,2&5) to capture non-instantaneous changes between important observations further differentiates our approach.
>
> 2. a) The reviewer asked why an average over the window size is a necessary part of the algorithm as described in the aggregation step (Section 3.3). The reason for this is that in order to extend beyond instantaneous attribution and to capture the varying importance of the same feature observation to predictions at different times, our method calculates the importance with respect to multiple predictions over a window of time steps. Aggregation is required to summarize this to a single importance score for each observation in the time series.
>     In terms of the choice of mean for aggregation, to justify this choice further we added Appendix A.3 to the updated manuscript, where we considered max as an alternate method of aggregation to mean. Empirical results show that mean outperforms max on the MIMIC-III Mortality task in AUC Drop. We believe aggregating with mean makes the most sense theoretically as it priortizes the case when the importance is maintained across multiple prediction steps and reduces the volatility of the score.
>
>     b) We thank the reviewer for for the comments on clarifying $i(S)_a^b$, and their suggestions for improving the clarity of Fig. 1b and Fig. 2. We have updated and reordered the discussion in Section 3.1. We have modified Fig. 1b to match the notation in Eq. 2 more clearly and added a reference below Eq. 2. For Fig. 2 we have added both the predictions without masking $P$ and with masking $Q$ used to compute $i(S)_a^b$.

---

> ### Author Response · Authors · 2022-11-16
> **Response to Reviewer CWz4 (2/2)**
>
> 3. Regarding the simplicity of the example illustrated in Figure 2 (which we believe the reviewer is referring to), we thank the reviewer for their comment and would like to highlight that Figure 2 is intended to be a toy example to provide intuition on how WinIT works in the simplest setting. We believe that adding multiple signal interactions to a toy illustration would detract from this goal. To demonstrate our method can effectively handle multiple signal patterns we include experiments on datasets with increasing complexity - in particular the simulated spike, delayed-spike and state datasets, and the real-world MIMIC-III dataset, in which we expect the predictions to be determined by multiple signals. We believe that the consistently strong performance of our method across these datasets provides convincing evidence of the usefulness of our approach.
>
> 4. Regarding the difference between the AUC Drop and Prediction Change in Fig. 4, we note that while the relative ordering of explainability approaches changes signficantly across masking methods, as explored in Section 5.3, for a given masking method the order of performance is relatively consistent across AUC Drop and Prediction Change. There are however some differences as these two metrics are chosen to illustrate different facets of the explanations. In particular AUC Drop depends on the order of the predictions rather than the size of the prediction change. Consider for example Dynamask with STD masking in Fig. 4, which has a similar Prediction Change to WinIT, but a lower AUC Drop. To explore this further, we conducted an additional analysis to compare the change in ranking of predictions before and after masking important features. We found that our method has a Kendall tau rank correlation of 0.454, while Dynamask has 0.705, suggesting that while their importance scores change the values of the predictions they do not drastically change their order, and hence have a lower AUC Drop.

---

### Official Review · Reviewer_4Fvg · 2022-10-24

**Confidence:** 3
**Correctness:** 4
**Technical Novelty And Significance:** 3
**Empirical Novelty And Significance:** 3
**Recommendation:** 8

**Clarity, Quality, Novelty And Reproducibility:**

The reporting is clear and good quality, the proposed approach is new, and the relevant literature has been properly and comprehensively cited. I did not find any mention of data and code availability, which may pose a severe bottleneck for reproducibility.

**Strength And Weaknesses:**

Strengths:
+ Addresses a relevant and timely issue of improving interpretability of machine learning techniques
+ Comprehensive set of evaluation approaches
+ The method shows clearly improved performance over alternatives

Weaknesses:
- Fixed window size is a potentially weak spot in the algorithm design as it will require heuristic window size selection schemes
- LImitations and weaknesses of the method could be addressed in more detail as these are relevant for any subsequent application
- Publicly available implementation does not appear to be available, limiting the practical utility.

**Summary Of The Paper:**

This paper provides a new method, WinIT, for interpreting predictions of multivariate time series. It provides an importance score for each feature in a multivariate time series according to explainability, considering the dependencies of features between time steps and variation in feature importance over time. The method considers differences between consecutive time points and measures importance over multiple time steps to quantify temporal variation in importance. The novelty is in combining these two aspects in the importance score. Analyses on synthetic and real data sets are used to demonstrate advantages and consistency compared to currently available alternatives and assess model evaluation.

**Summary Of The Review:**

Overall, this is am interesting and well documented contribution on a timely topic. Reproducibility of the work is a key limitation.

---

> ### Author Response · Authors · 2022-11-16
> **Response to Reviewer 4Fvg**
>
> We thank the reviewer for their insightful comments and constructive suggestions. We value their assessment that our approach is new and provides clear improvement over alternatives. We address each point raised by the reviewer below and have uploaded a revised version of our manuscript.
>
> 1. We agree that it is helpful to include further discussion of the limitations of our method to support subsequent work and have added this to our Conclusions (Section 6).
>
>     One area we have highlighted is to extend the time series explainability beyond saliency maps, to capture higher level patterns. We postulate that some signals that lead to prediction changes would satisfy a "pattern". For example, if feature 1 continues to increase significantly for 5 time steps, the prediction will suddenly jump higher. An explanation by saliency map would highlight the 5 time steps of feature 1 at best, but fail to provide meaningful explanations about the “pattern” of the features over time. A better explanation would be "feature 1 has been increasing over the past 5 time steps", which extends beyond saliency map based explanations. This argument can also apply to patterns across features as well. Thus we are excited to explore different formats of explaining time series data. We leave this to future work.
>
> 2. We will make the implementation of our method publicly available on a GitHub repository with a final version of this draft. We have added a reproducibility statement to the manuscript (see page 10), and will update both with a link to our repository.

---

### Official Review · Reviewer_vrY7 · 2022-10-25

**Confidence:** 3
**Correctness:** 4
**Technical Novelty And Significance:** 3
**Empirical Novelty And Significance:** 3
**Recommendation:** 8

**Clarity, Quality, Novelty And Reproducibility:**

The paper is very novel, to the best of my knowledge. It is also relative clear even though at places, it can be improved.

**Strength And Weaknesses:**

Strengths:
1-	Addresses a very important problem
2-	Novelty
3-	Extensive experimental study

Weakness:
1-	Notation could be improved
2-	At places, it is difficult to follow the text.


**Summary Of The Paper:**

This paper introduces a new explainability framework for multi-variate time series data. This new explainability framework is a feature removal model. The authors motivate their approach by the fact that the observation of same feature over subsequent time steps is not independent and that feature can have varying level of importance to the model prediction over time. Using extensive experimental study, the authors show that the proposed method is superior to the existing explainability models for time series data.

**Summary Of The Review:**

This paper addresses the practically important problem of explainability of black box classifiers for time series data. The authors introduce a novel framework that calculates the feature importance as the difference of the importance over multiple time steps and aggregates
the importance of each observation over a window of predictions. The authors also a generative approach for the feature removal, which is interesting. The authors also provide extensive experimental studies to support that the new framework works better than the existing explainability approaches for time series data. Overall, this is a good paper.

---

> ### Author Response · Authors · 2022-11-16
> **Response to Reviewer vrY7**
>
> We thank the reviewer for their valuable feedback on our manuscript. We especially appreciate their comments on the novelty of our work, the importance of our topic, and our extensive experimental study.
>
> To improve the clarity of the notation we have added a table to the Appendix summarizing the notation used throughout the paper (see Table 4 Appendix A.1).
>
> In addition, to improve the readability of the text we have re-ordered Section 3.1, the Importance Score Formulation, which utilizes significant notation and improved Fig. 1b and Fig. 2 to help make this section easier to follow.

---

### Author Response · Authors · 2022-11-16
**General Response**

We thank all the reviewers for their time and valuable feedback. We would like to address common feedback in this general response.

Regarding the window size selection, while it is true that the window size is an extra parameter to set, we think that this is not a significant limitation for several reasons. Firstly, since this parameter is interpretable - namely, it represents the longest range signal that will impact the predictions, it can be informed by domain experts and the task in consideration. Secondly, beyond a threshold, our method is not sensitive to the choice of window size. For example, as we see from the MIMIC-III mortality experiment in Figure 6(b) the AUC drop stabilizes after N=10, suggesting that there are a wide range of choices of window size for which the explainability performance is very similar.

For clarity and readability of the paper, the following changes have been made to the manuscript and a new version uploaded.
- We have refined and reordered the paragraphs in Section 3.1 to make the importance score formulation more readable.
- We improved Figure 1 and 2 to make it more understandable.
- We added limitations and future work to our conclusion section.
- We added a reproducibility statement, where a link to a publicly available implementation of our code will be added.
- We added summary table of the notation used throughout the paper in the appendix.
- We added a note on the aggregation method described in Section 3.3 in the appendix.
- Minor change of wordings and reformatting for conciseness.

---

### Decision · Program_Chairs · 2023-01-20

**Decision:**

Accept: poster

**Justification For Why Not Higher Score:**

The paper may benefit from better justifications of the proposed algorithm.

**Justification For Why Not Lower Score:**

The proposed important score is novel and demonstrated to be superior to alternatives.

**Metareview: Summary, Strengths And Weaknesses:**

This paper introduces a new explainability framework for interpreting predictions of multivariate time series. It provides an importance score for each feature in a multivariate time series according to explainability, taking into account the dependencies of features between time steps and variation in feature importance over time. The method considers differences between consecutive time points and measures importance over multiple time steps to quantify temporal variation in importance. The proposed important score is novel and demonstrated to be superior to alternatives. The paper may benefit from better justifications of the proposed algorithm.

**Note From Pc:**

if the above contains the word "oral" or "spotlight" please see: "oral" presentation means -> notable-top-5% and "spotlight" means -> notable-top-25%. As stated in our emails, we are disassociating presentation type from AC recommendations